# Discovery of New Triterpenoids Extracted from *Camellia oleifera* Seed Cake and the Molecular Mechanism Underlying Their Antitumor Activity

**DOI:** 10.3390/antiox12010007

**Published:** 2022-12-21

**Authors:** Zelong Wu, Xiaofeng Tan, Junqin Zhou, Jun Yuan, Guliang Yang, Ze Li, Hongxu Long, Yuhang Yi, Chenghao Lv, Chaoxi Zeng, Si Qin

**Affiliations:** 1The Key Laboratory of Cultivation and Protection for Non-Wood Forest Trees, Ministry of Education, Central South University of Forestry and Technology, Changsha 410004, China; 2School of Economics and Management, Hunan Open University, Changsha 410004, China; 3Key Laboratory of Non-Wood Forest Products of State Forestry Administration, College of Forestry, Central South University of Forestry and Technology, Changsha 410004, China; 4National Engineering Laboratory for Rice and Byproducts Processing, Food Science and Engineering College, Central South University of Forestry and Technology, Changsha 410004, China; 5Laboratory of Food Function and Nutrigenomics, College of Food Science and Technology, Hunan Agricultural University, Changsha 410128, China

**Keywords:** *Camellia oleifera* seed cakes, theasaponin derivatives, identification, molecular mechanism, antitumor activity

## Abstract

Theasaponin derivatives, which are reported to exert antitumor activity, have been widely reported to exist in edible plants, including in the seed cake of *Camellia oleifera* (*C.*), which is extensively grown in south of China. The purpose of this study was to isolate new theasaponin derivatives from *C.* seed cake and explore their potential antitumor activity and their underlying molecular mechanism. In the present study, we first isolated and identified four theasaponin derivatives (compounds **1**, **2**, **3,** and **4**) from the total aglycone extract of the seed cake of *Camellia oleifera* by utilizing a combination of pre-acid-hydrolysis treatment and activity-guided isolation. Among them, compound **1** (C**1**) and compound **4** (C**4**) are newly discovered theasaponins that have not been reported before. The structures of these two new compounds were characterized based on comprehensive 1D and 2D NMR spectroscopy and high-resolution mass spectrometry, as well as data reported in the literature. Secondly, the cytotoxicity and antitumor property of the above four purified compounds were evaluated in selected typical tumor cell lines, Huh-7, HepG2, Hela, A549, and SGC7901, and the results showed that the ED_50_ value of C4 ranges from 1.5 to 11.3 µM, which is comparable to that of cisplatinum (CDDP) in these five cell lines, indicating that C4 has the most powerful antitumor activity among them. Finally, a preliminary mechanistic investigation was performed to uncover the molecular mechanism underlying the antitumor property of C**4**, and the results suggested that C**4** may trigger apoptosis through the Bcl-2/Caspase-3 and JAK2/STAT3 pathways, and stimulate cell proliferation via the NF-κB/iNOS/COX-2 pathway. Moreover, it was surprising to find that C**4** can inhibit the Nrf2/HO-1 pathway, which indicates that C**4** has the potency to overcome the resistance to cancer drugs. Therefore, C**1** and C**4** are two newly identified theasaponin derivatives with antitumor activity from the seed cake of *Camellia oleifera*, and C**4** is a promising antitumor candidate not only for its powerful antitumor activity, but also for its ability to function as an Nrf2 inhibitor to enhance the anticancer drug sensitivity.

## 1. Introduction

*Camellia* is an economically and phylogenetically important member of the Theaceae family. There are more than 300 Camellia species widely distributed throughout China, Japan, and the tropical and subtropical regions of Asia [1,2]. Several *Camellia* species have been domestically cultivated, and numerous cultivars have been generated to satisfy the increasing demand for them, as they are used as the raw material to produce popular commercial products, including health-promoting tea ingredients [3], horticulturally showy flowers, and high-quality edible oils [4,5]. Because of its high nutritional and economic value, *C*. *oleifera* Abel. is considered to have the highest production value among the *Camellia* species [6]. This small perennial tree has been cultivated and utilized for more than 2000 years in China, and it is also one of the four major woody oil plants in the world [7]. In the food and cosmetic industries, *C*. *oleifera* seeds are primarily used to produce a pure natural edible oil [8,9]. It is recommended by the Food and Agriculture Organization of the United Nations (FAO) as a high-quality and healthy vegetable oil because it is rich in unsaponifiable components, such as sterols, fatty alcohols, and tocopherols [10].

*C*. *oleifera* seed cake (Abbreviated as *camellia* seed cake) is the residual byproduct of the tea oil extraction from *C*. *oleifera* seeds [7]. Although it constitutes 80% of the total seed mass, the oil cake is either burned or discarded after oil extraction, which is an enormous waste of organic resources [11]. Full exploitation of the potential value of *Camellia* seed cake will not only generate economic value but will also prevent air, water, or soil pollution. Saponins are the most important components in *Camellia* seed cake, accounting for about 15% to 20% by weight [12]. More than 30 types of saponins have been characterized from *Camellia* seed cake, and new saponins continue to be isolated and identified. Many of them have promising cytotoxicity in human cancer cell lines [13,14,15]. However, previous studies often did not screen the active components according to the functional activity of the target compounds, resulting in a low screening efficiency, and there are few studies on the antitumor function-mechanism of theasaponin derivatives. Therefore, in order to rapidly discover new saponin resources with antitumor activity from *Camellia* oleifera seed cake, the combination of a high-efficiency pretreatment with an activity-guided isolation is a potential and necessary strategy.

From the cytotoxic screening test, we found that the acid-treatment fraction had better cytotoxic activities in the tested tumor cell lines compared to the total saponins fraction (Appendix A). Herein, we report the utilization of a new isolation strategy, which is a combination of pre-acid-hydrolysis treatment and activity-guided isolation. By performing this strategy, we discovered four theasaponin derivatives, including two new ones (**1** and **4**), as well as their in vitro cytotoxic activities against five human tumor cell lines, namely Huh-7, HepG2, Hela, A549, and SGC7901. The mechanism of action of the active compounds was also investigated. The purpose of this study was to isolate new theasaponin derivatives from *C*. seed cake and to explore their potential antitumor activity and their underlying molecular mechanisms.

## 2. Materials and Methods

### 2.1. General Experimental Procedures

The UV spectra were recorded by a PERSEE UV-VIS spectrophotometer T9 (Beijing, China). The IR spectra were recorded by a Thermo Nicolet Nexus 470 FT-IR spectrometer (Thermo Fisher Scientific, Waltham, MA, USA). NMR spectra were recorded by a Bruker AVANCE III 400 NMR spectrometer (Bruker, Germany), using CDCl_3_ as the solvent, and the chemical shifts were referenced to the solvent residual peak. HRESIMS data were acquired using Agilent 6250 TOF LC/MS system. Silica gel (200–300 mesh, Qingdao Marine Chemical, Co. Ltd., Qingdao, China) and AB-8 macroporous adsorption resin (Solarbio Life Science Co., Ltd., Beijing, China) were used for open column chromatography (CC). XBridge Shield RP18 HPLC column (3.5 μm 4.6 ×150 mm) and Waters Acouity HPLC system were used for HPLC analysis. Semipreparative RP-HPLC was performed on a Waters 2695-2489 system with a GLP-ID (150 mm × 450 mm) preparative column (Chengdu Gelai Co. Ltd., Chengdu, China). TLC analyses were carried out on pre-coated silica gel GF254 plates (Qingdao Marine Chemical Co. Ltd., Qingdao, China). All of the solvents were analytical grade.

### 2.2. Sample Preparation

The seeds of tea plant *Camellia oleifera* Abel. were collected from She-Jiang Town, Mei-Zhou City, Guang-Dong Province, China in 2020. The *Camellia* seed cake was generated by squeezing out the edible tea oil and then used for the extraction.

### 2.3. Extraction, Acid Hydrolysis, and Isolation

Five kilograms of *Camellia* seed cake were ground into powder and refluxed with petroleum ether for 2 h. The defatted seed cake was dried and further extracted with 75% EtOH (Shanghai Yuanye Bio-Technology Co., Ltd., Shanghai, China) three times. The crude extract was then loaded into an AB-8 macroporous resin column and eluted with 1% aqueous NaOH followed by a sequential mixture of methanol–H_2_O (0%, 10%, 30%, 80%). The fraction eluted with 80% methanol was concentrated under reduced pressure to achieve a total saponin fraction (251 g).

The solution of the total saponin dissolved in 3N HCl (methanol–H_2_O, 1:1) was refluxed for 5 h. The reaction mixture was neutralized with 10% aqueous NaOH, followed by EtOAc extraction three times. The EtOAc layer was then dried, and a total of 40 g of aglycone extracts were harvested from *Camellia* seed cake.

The total aglycone extracts were loaded into a silica gel column and eluted with petroleum ether-EtOAc (30:1–1:1, gradient system) to generate two fractions designated as Fr. A (0.8 g) and Fr. B (1.4 g). Fr. A was further purified by semipreparative RP-HPLC using a mobile phase of MeCN-H_2_O (70:30, *v/v*) to provide fraction C**3** (231 mg) and fraction C**4** (70 mg). Separation of the Fr. B was performed using semipreparative RP-HPLC (isocratic 85% MeCN in H_2_O), which yielded fraction C**1** (52 mg) and fraction C**2** (234 mg).

### 2.4. Cytotoxicity Bioassay

Cytotoxicity of these isolates was evaluated in five human cancer cell lines: Huh-7, HepG2, Hela, A549, and SGC7901. All cell lines were provided by Procell Life Science & Technology Co., Ltd. (Wuhan, China). The viability of the cells after isolates treatment was tested using the MTT assay. Three independent experiments were done for each treatment.

Experimental reagent configuration method: (1) Electrophoresis solution: 15.1 g Tris (BioFroxx, Hessen, Germany), 94 g Glycine (BioFroxx, Germany), and 5 g SDS (BioFroxx, Germany) were weighed by analytical balance and weighing paper, respectively, and placed into beaker together. The reagent was first dissolved by stirring with a certain volume of ultrapure water. After dissolving, the corresponding solution was poured into a 1000 mL volumetric flask, and ultrapure water was added for constant volume. After the volume was fixed, it was put into a washed blue-capped bottle and transferred to the refrigerator at 4 °C for further storage. (2) Lysate: 5 mL RIPA (Beyotime Biotechnology Co. Ltd., Nantong, China) lysate was sucked with a pipette gun and placed in a 15 mL centrifuge tube, and the centrifuge tube was placed on the ice box prepared in preparation. Subsequently, 50 μL of PMSF (Beyotime, China) was added to it according to the proportion, mixed well with a shaker, and transferred to the refrigerator at 4 °C for storage. (3) Antibody: 3–5 mL of antibody was prepared and put into the labeled antibody incubation box, and then transferred to the refrigerator for storage at 4 °C. The secondary antibody of the corresponding species was diluted with TBST (Solarbio, Beijing, China) at a ratio of 1:10,000. Generally, it can be reused 2–3 times.

The cells were cultured in DMEM (Gibco, New York, NY, USA) containing 10% fetal bovine serum and 1% double antibody, and the cells were used when the cell density reached 80–90% of the culture flask. Next, 5 mL of PBS (Gibco, USA) was added to clean the remaining culture in the culture flask, and the PBS (Gibco, USA) was aspirated clean. Then, 350 mL of trypsin was added, and the culture flask was quickly transferred to an incubator containing 5% CO_2_ at 37 °C. We paid attention to observe when the cells became slightly round, and quickly added 2 mL DMEM to finish the culture to terminate digestion. The cells were transferred to a 15 mL centrifuge tube, centrifuged at 1500 RPM for 5 min, and resuspended with fresh culture. The cell suspension was diluted to 100,000 cells per 1mL according to the purpose in a hemacyte counter plate, 100 μL per well in a 96-well plate, and at least three multiple wells per group. The cells were incubated at 37 °C and 5% CO_2_ for 24 h before adding drugs. After incubation for 48 h, 10% CCK8 was added and incubated for 1–3 h. The absorbance was measured by microplate reader at 450 nm. The samples were loaded in the order of compound concentration, 6.25 mg/mL, 1.56 mg/mL, 0.39 mg/mL, and blank, including 2 wells of compound sample and 4 wells of blank sample. Each well was loaded with 20 μL, and electrophoresis was carried out at 140 V for 70 min. The 0.22 μm PVDF membrane was electrotransferred for 60 min at 100 V voltage. The blocking solution was 5% skim milk powder at 37 °C for 1.5 h with shaking. Primary antibody STAT3 (Abcam, MA, USA) was 1:2000, and primary antibody NFKBP65 (Abcam, USA), ERK1(Abcam, USA) and α-tublin (Abcam, MA, USA) were diluted to 1:1000 and incubated overnight at 4 °C with shaking. After washing the PVDF membrane, HRP (Abcam, USA) was diluted to 1:500 and incubated for 1 h at 25 °C with shaking. After cleaning the PVDF membrane, HRP luminescent solution was added and photographed.

### 2.5. Cell Apoptosis Assay

Cells were cultured in 12-well culture plates with different treatments for 4 h, then washed twice with cold PBS (Beyotime Biotechnology Co. Ltd., Nantong, China), according to the instruction of the Annexin V-FITC/PI kit (Beyotime Biotechnology Co. Ltd., China), resuspended in Annexin binding buffer (10 mM HEPES, 140 mM NaCl, 2.5 mM CaCl_2_ and pH 7.4) to a concentration of 1 × 10^6^ cells/mL. Next, 100 μL of the resuspended cells was transferred to 5 mL tube, 5 μL of Annexin V-FITC and 5 μL PI were added, and the cells were gently vortexed. After 15 min of incubation at room temperature, 400 μL of binding buffer was added to each tube and cells were analyzed by flow cytometry within 30 min.

### 2.6. Cell Counterstaining

Cells were cultured in DMEM medium containing 10% FBS (Beyotime Biotechnology Co. Ltd., Nantong, China) and 1% double antibiotics (100 U/mL penicillin, 100 mg/mL streptomycin, Gibco, USA), then washed 5 mL PBS. Next, we added 350 mL pancreatic enzyme and quickly transferred the culture flask to an incubator containing 5% CO_2_ at 37 °C. After cell rounding, we quickly added 2 mL of culture solution to stop digestion. The cells were transferred to a 15 mL centrifuge tube, centrifuged at 1500 RPM for 5 min, and resuspended with fresh culture. The cell suspension was diluted to 1 × 10^6^ cells/mL and incubated at 37 °C and 5% CO_2_ for 24 h, and the compound was added. After incubation for 48 h, we added 100 μL of calcein (Calcein AM/PI), and incubated for 30 min, and then performed image analysis with a high connotation cell imaging analysis system. Six transcription factors were selected. GAPDH or Actin was used as a reference gene.

### 2.7. Western Blot Analysis

Western blot analysis was performed as described previously [16]. In brief, A549 U251 PAN02 and HepG2 cells (1 × 10^6^ cells/mL) were pre-cultured in 6-cm dish for 24 h, and starved in serum-free medium for another 2.5 h to eliminate the influence of FBS. Then the cells were treated with or without MV for 2 h before exposure to C**3** or C**4** for different time periods. The harvested cells were lysed and the supernatants were boiled for 5 min. Protein concentration was determined by using a dye-binding protein assay kit (Beyotime Institute of Biotechnology) according to the manufacturer’s manual. Equal amounts of lysate protein were subject to 10% SDS-PAGE and electrophoretically transferred onto a PVDF membrane (Amershan Pharmacia Biotech, Little Chalfont, UK). After being blocked, the membrane was incubated with the specific primary antibody (mTOR, p-mTOR, STAT3, p-STAT3, Bcl-2, Bax, GAPDH, β-actin, JAK2, β-catenin, caspase-3, TNF-α, Nrf2, HO-1,NF-κB, Inos, COX-2; Abcam, MA, USA) overnight at 4 °C, and further incubated for 1 h with HRP-conjugated secondary antibody. Anti-rabbit and anti-mouse were purchased from Abcam, USA. Bound antibodies were detected using ECL system with Lumi Vision PRO machine (TAITEC, Saitama, Japan). The relative amount of proteins associated with a specific antibody was quantified using the Lumi Vision Imager software.

### 2.8. Statistical Analysis

All data were analyzed by SPSS Software. All samples were measured in triplicate and data were expressed as the mean ± SD from the three different experiments. One-way ANOVA was used to measure statis tical differences between the means within each experiment. *p* < 0.05 was considered as significant statistical difference.

## 3. Results

### 3.1. Extraction, Identification, and Characterization of Theasaponin Derivatives from Camellia oleifera

Considering the chemical construction of saponins, we conducted a pre-acid-treatment of the total saponins fraction of *Camellia* seed cake. The screening test results showed that the acid-treatment fraction exhibited better cytotoxic activity in most of the cancer cell lines (Appendix A). Then, *Camellia* seed cake was extracted with 75% methanol, eluted through silica gel column and finally detected by semipreparative RP-HPLC. At last, four major compounds were separated and identified, compounds **1**, **2**, **3,** and **4** (C**1**, C**2**, C**3**, and C**4**), as shown in Figure 1. Among them, two new theasaponin derivatives, C**1** and C**4**, were reported for the first time to exist in *Camellia oleifera*.

To verify the exact chemical structure of the above four compounds, ^1^H and ^13^C NMR were performed. The results showed that C**1** was isolated into a white amorphous powder. The positive-ion HR-ESI–MS spectrum of C**1** displayed an [M + H]^+^ peak at *m/z* 571.4025 (calcd. 571.3999), corresponding to a molecular formula of C35H54O6. The IR spectrum indicated the presence of a hydroxyl group (3525 cm^−1^) and an *α*, *β*-unsaturated ester group (1710 and 1650 cm^−1^). The ^1^H NMR (Table 1) spectrum of C1 indicated a formyl proton at *δ*_H_ 9.37 (1H, s, H-23), an aromatic singlet at *δ*_H_ 5.31 (1H, t, *J* = 3.9 Hz, H-12), two isolated oxygenated methylene proton signals at *δ*_H_ 3.64 (1H, d, *J* = 11.3 Hz, H-28) and 3.38 (1H, d, *J* = 11.3 Hz, H-28), three oxygenated methine protons at *δ*_H_ 3.75 (1H, dd, *J* = 11.4, 4.6 Hz, H-3), 5.80 (1H, brs, H-16), and 3.89 (1H, dd, *J* = 12.8, 5.4 Hz, H-22), six methyl proton signals at *δ*_H_ 1.05 (3H, s, H-24), 0.98 (3H, s, H-25), 0.93 (3H, s, H-26), 1.40 (3H, s, H-27), 0.85 (3H, s, H-29), and 0.93 (3H, s, H-30), as well as resonances for protons of an angeloyl group [*δ*_H_ 6.91 (1H, q, *J* = 6.4 Hz, H-3′), 1.82 (3H, d, *J* = 7.2 Hz, H-4′), and 1.87 (3H, s, H-5′)]. The ^13^C NMR (Table 1) spectrum displayed the resonances of 35 carbons, including two carbonyl carbons (*δ*_C_ 207.3 and 169.5), four olefinic carbons (*δ*_C_ 141.4, 139.2, 123.7, and 129.1), four oxygenated carbons (*δ*_C_ 76.6, 73.2, 72.0, and 71.5), eight methyls (33.1, 27.3, 24.9, 16.7, 16.0, 14.7, 12.5, and 9.1), eight methylenes (*δ*_C_ 46.4, 44.1, 38.4, 32.1, 31.8, 26.2, 23.5, and 20.8), three methines (*δ*_C_ 48.4, 46.8, and 42.0) and six quaternary carbons (*δ*_C_ 55.3, 43.9, 41.4, 40.3, 36.0, and 31.6), supported by the HSQC results. These data suggested that C1 contains an oleanane structure in the molecule, which is in accordance with the other reports [14,15]. In the HMBC experiment, the correlations from H-24 to C-3/C-4/C-5/C-23, H-25 to C-1/C-5/C-9/C-10, H-26 to C-7/C-8/C-9/C-14, H-27 to C-8/C-13/C-14/C-15, and H-29/H-30 to C-19/C-20/C-21 further confirmed the existence of an oleanane skeleton. The HMBC correlations between H-1/H-2/H-24 and C-3 (*δ*_C_ 72.0), H-18/H-21/H-28 and C-22 (*δ*_C_ 76.6), and H-16/H-18/H-22 and C-28 (*δ*_C_ 73.2) were used to place the three hydroxy groups at C-3, C-22, and C-28, respectively (Figure 2). The presence of an angeloyl substituent group was further supported by HMBC correlations between H-3′/H-5′ and C-1′, H-3′/H-4′/H-5′ and C-2′, H-4′/H-5′ and C-3′, as well as the NOESY correlations between H-4′ and H-5′ (Figure 2). The position of the angeloyl group at C-16 could be clarified based on HMBC correlations between H-16 and C-1′. The relative configuration of C**1** was determined from the NOESY experiment (Figure 2), and its structure was characterized in Figure 1.

The molecular formula of C4 was predicted to be C_47_H_70_O_14_ based on its ^13^C NMR and HRESIMS data (*m*/*z* 859.4821 [M + H]+, calculated for C_47_H_71_O_14_, 859.4844), indicating 13 indices of hydrogen deficiency. The IR spectrum of C4 had absorption bands at 1708 and 1646 cm^−1^ representing a carboxyl and *α, β*-unsaturated ester, and broad bands at 3447 and 1044 cm^−1^ suggesting a glycosidic structure [14,15]. These data was further supported by the NMR spectra results (Table 1). Detailed analysis of the 1D and 2D NMR data suggested that the structure of C4 was similar to that of compound 1. The significant difference was the additional signals for the glycosidic structure [*δ*_H_ 4.27 (1H, d, *J* = 7.5 Hz, H-a), 3.33 (1H, m, H-b), 3.50 (1H, m, H-c), 3.66 (1H, m, H-d), 3.87 (1H, m, H-e), 3.80 (3H, s, H-f-Me); *δ*_C_ 104.1 (C-a), 73.3 (C-b), 75.5 (C-c), 71.4 (C-d), 71.7 (C-e), 169.9 (C-f), 52.9 (C-f-Me)] and two angeloyl substituent groups in C**4**. The glycosidic structure was connected to C-3 of the oleanane skeleton demonstrated by the HMBC correlations of H-a and C-3 as well as H-3 and C-a (Figure 2). The two angeloyl groups could be located at C-21 and C-22, respectively, based on the HMBC correlations of H-21 and C-1′, H-22 and C-1″. Hence, the structure of C**4** was established as shown in Figure 1.

C**2** and C**3** were both obtained as white amorphous powders, and analysis of their MS and NMR data (Table 1) suggested that they were both oleanane triterpenes and had a closely theasaponin skeleton. Compared with compound **1**, the NMR spectroscopic data of compound 2 and 3 indicated the presence of an angeloyl group [*δ*_H_ 6.93 (1H, qd, *J* = 7.0, 1.1 Hz, H-3′), 1.82 (3H, dd, *J* = 7.0, 1.1 Hz, H-4′), and 1.87 (3H, s, H-5′), *δ*_H_ 169.6 (C-1′), 129.1 (C-2′), 139.2 (C-3′), 14.7 (C-4′), 12.5 (C-5′) for 2; *δ*_C_ 6.11 (1H, qd, *J* = 7.2, 1.4 Hz, H-3′), 1.99 (3H, dd, *J* = 7.2, 1.4 Hz, H-4′), and 1.89 (3H, t, *J* = 1.4, H-5′), *δ*_C_ 169.0 (C-1′), 127.7 (C-2′), 139.2 (C-3′), 16.0 (C-4′), 20.8 (C-5′) for 3]. In compound 2, the HMBC correlation between H-16 and C-1′ was used to locate the angeloyl substituent group at C-16. In combination with the HMBC correlations, compound 2 was identified as camelliagenin A 16-tiglate. Likewise, the structure of compound 3 was determined as camelliagenin A 22-tiglate due to the HMBC correlations between H-22 and C-1′. Although 2 and 3 are known compounds that have been previously characterized elsewhere, their ^1^H and ^13^C NMR data were fully disclosed for the first time (Table 1).

### 3.2. Antitumor Activity of the Identified Theasaponin Derivatives in Cancer Cell Lines

All the isolates were evaluated for their cytotoxicities in five human tumor cell lines (Human hepatocellular carcinoma cell Huh-7 and HepG2, human cervical carcinoma cell Hela, human lung carcinoma cell A549, and human gastric carcinoma cell SGC7901) using MTT assay. The widely used anticarcinogen cisplatinum (DDP) was used as the positive control. As shown in Table 2, all of the isolated theasaponin derivatives exhibited significant cytotoxic activity in the human tumor cell lines, especially C**4**. Compound **4** showed potent cytotoxicity, with an ED_50_ ranging from 1.5 to 11.3 µM, which was compared with DDP.

According to the above results of the preliminary structure relationship analysis, we noticed that an additional aldehyde group at the C-23 position in C**1** could increase the cytotoxic activity in comparison with C**2**. Therefore, we hypothesized that the C-23 position could be a promising site of theasaponin scaffolds.

### 3.3. Effects of C**3** and C**4** on Viability of Three Typical Cancer Cell Lines

Considering the cytotoxic activities of the tested theasaponin derivatives and the importance of C**3** and C**4** as potential promising antitumor compounds, we further investigated the effects of C**3** and C**4** on the expressions of proteins belonging to the signaling pathways related to survival, migration, and invasion in cells, such as the JAK2/STAT3, β-catenin-mTOR, Bcl-2/Bax-Caspase 3, and NF-κB/Nrf2 pathways.

According to the results of calcein AM/PI staining, we discovered significant apoptotic effects for the A549, U251, and PAN02 cells in C**3** and C**4** (Figure 3A,C,E). In three cell models, lots of orange fluorescence representing dead cells was observed in the C**3** and C**4** groups, which were compared with the control. The cell mortality of the C**4** group was higher than that of the C**3** group in A549 and PAN02 cells (Figure 3A,C). However, the cell death rate of C**3** group was greater than that of C**4** group in U251 cells (Figure 3B). These results indicated that C**3** and C**4** significantly inhibited the proliferation of cancer cells, and that C**4** is more effective than C**3**.

The expressions of C**3** and C**4** in proteins associated with apoptosis in three kinds of cells are shown in Figure 3B,D,F. Compounds **3** and **4** promoted apoptosis by regulating different proteins in these cells. C**3** did so by up-regulating Bax, and C**4** by down-regulating p-STAT3 and Bcl-2 in A549 cells (Figure 3B). In U251 cells, C**3** promoted apoptosis by down-regulating p-mTOR and p-STAT3, and C**4** by down-regulating p-STAT3 and Bcl-2 (Figure 3D). Both compounds **3** and **4** promoted apoptosis by down-regulating p-mTOR, p-STAT3, and Bcl-2 in PAN02 cells (Figure 3F). The results illustrated that both C**3** and C**4** had obvious effects on promoting apoptosis. At the same time, the effect of C**4** is better than that of C**3**. C**4** considerably reduced the expression levels of p-STAT3/STAT3 in A549, U251, and PAN02 cells, compared with control cells. Meanwhile, it could also attenuate the anti-apoptotic protein Bcl-2 and reduce the Bcl-2/Bax ratio.

### 3.4. Effect of C**4** on Apoptosis of HepG2 Cells

C**4** was selected to confirm its antitumor effect because it is more proapoptotic than C**3**. At the time of apoptosis, the cell volume becomes smaller, so the forward scattered light is reduced, which is often considered to be one of the characteristics of apoptotic cells. In addition, chromosome degradation, nuclear rupture, and the increase of intracellular particles occur in apoptotic cells, so the side scattered light of apoptotic cells often increases. According to the results of flow cytometry (Figure 4A), the apoptosis rate of C**4** was definitely enhanced in a dose-dependent manner. In the early apoptotic cells (FITC+/PI−, Q4), the value of Blank group was 9.32, and the values C**4** (0.78125 μM) and C**4** (1.5625 μM) were 10.7 and 9.73, respectively. This suggests that C**4** can significantly promote early apoptosis. For late apoptotic cells (FITC+/PI+, Q2), the value of Blank group was 9.63, and the value of C**4** (0.78125 μM) was 12.5, which indicated that C**4** (12.5 μM) could promote cell apoptosis. In addition, C**4** promoted late cell apoptosis in a dose-dependent manner. To obtain more accurate molecular evidence of C**4**′s antitumor activity, typical apoptosis-related proteins were detected in HepG2 cells. The results in Figure 4B indicated that the expression of Caspase-3 and TNF-α had significantly declined, compared with blank, which promoted the cell apoptosis. Simultaneously, compared with blank, the expression of β-catenin was considerably enhanced to promote the apoptosis. However, JAK2 did not change significantly. These results demonstrate that C**4** promotes apoptosis by down-regulating β-catenin and up-regulating Caspase-3 and TNF-α.

### 3.5. Effect of C**4** on Redox Related Nrf2 and NF-κB Pathways in LPS-Induced HepG2 Cells

To elucidate the molecular mechanism of C**4**′s antitumor activity, typical apoptosis-related proteins were detected in LPS-induced HepG2 cells. Nrf2 and its downstream protein HO-1 were tested. Inflammatory factors NF-κB, iNOS, and COX-2 were detected. As shown in Figure 5, the LPS group was significant increased compared with blank. This indicates the inflammatory model has been successfully established. As presented in Figure 5B,C, the expression of Nrf2 and HO-1 was decreased considerably, both in the 3.125 and the 6.250 group, compared with the LPS group. Therefore, we hypothesized that C**4** might be a suitable inhibitor of Nrf2, which could enable antitumor drugs to exert synergistic effects in cancer cells. The expressions of NF-κB, iNOS, and COX-2 were enhanced remarkably in the 6.250 group, and increased, though not significantly, in the 3.125 group (Figure 5C–E). Based on these findings, C**4** may be an appropriate inflammatory inducer that is available to regulate proliferation.

Taken together, the above results indicated that C**4** may trigger apoptosis through the Bcl-2/Caspase 3 and JAK2/STAT3 pathways and stimulate cell proliferation via NF-κB/iNOS/COX-2 pathway. What is even more important is that C**4** can improve the resistance to cancer drugs by inhibiting the Nrf2/HO-1 pathway.

## 4. Conclusions and Discussion

In the present study, four cytotoxic theasaponin derivatives were reported, including two new ones (C**1** and C**4**), which were extracted from *Camellia* seed cake by the combination of pre-acid-hydrolysis treatment and activity-guided isolation. In addtion, molecular evidence shows that C**4** has excellent antitumor ability, which can significantly inhibit inflammatory pathways and targets as well as regulating the intracellular redox balance.

Studies have shown that *Camellia* seeds can reduce blood glucose and antioxidant effects and are able to slightly ameliorate carbon tetrachloride induced hepatotoxicity in rats by regulating inflammation [7,17,18,19]. *Camellia* seed cake is a byproduct of oil extraction and is generally discarded [20,21] Therefore, *Camellia* seed cake, as a plant with multiple active functions, has great research value.

It has been reported that the main components of *Camellia* seed have splendid antitumor activity [22,23]. Likewise, in the present study, the cytotoxicity and antitumor property of the above four purified compounds were evaluated in selected typical tumor cell lines, Huh-7, HepG2, Hela, A549, and SGC7901, and the results showed that the ED50 value of C**4** ranges from 1.5 to 11.3 µM, which is comparable to that of cisplatinum (CDDP) in these five cell lines, indicating that C**4** has the most powerful antitumor activity among them. The results indicated that C**4** may trigger apoptosis through the Bcl-2/Caspase 3 and JAK2/STAT3 pathways, and that, more importantly, C**4** can help overcome the resistance to cancer drugs by inhibiting the Nrf2/HO-1 pathway. Nrf2 is known as a master regulator of the redox balance. Nrf2 plays an important role in both tumor chemoprevention and tumor drug resistance, and some cancer cells can use this response to protect themselves from the damaging oxidative byproducts of abnormal metabolic activity and uncontrolled growth. Thus Nrf2 is traditionally considered to be a tumor regulator and an important indicator for the development of anticancer drugs. Our study indicates that C**4** is a promising mediator for cancer chemoprevention.

In the present study, Compound **3** worked by down-regulating p-mTOR and p-STAT3, and C**4** worked by down-regulating p-STAT3 and Bcl-2 in U251 cells. Compound **3** and Compound **4** both down-regulated p-mTOR, p-STAT3, and Bcl-2 in PAN02 cells. Cell injury activates the mTOR/STAT3 signaling pathway, thereby reducing cell apoptosis, which means that mTOR is an important signal transduction molecule. When mTOR is activated by external stimuli, the expression and activity of the downstream signaling molecules in the pathway will be increased, which can not only promote cell growth, proliferation, and differentiation, but can also regulate cell apoptosis. STAT3 can be activated by many cytokines and growth factors and is readily phosphorylated by the activated mTOR kinase. Activated STAT3 plays an important role in the control of cell growth, proliferation, differentiation, and apoptosis. Therefore, mTOR and STAT3 have crosstalk. In the future, we will conduct further studies to examine the effects of these components on the mTOR/STAT3 signaling pathway by using appropriate inhibitors, siRNA, or similar approaches.

The development of cancer is also closely related to inflammation. For example, inflammation-related DNA damage in cancer stem cell-like cells leads to the development of cancers with aggressive clinical features [24]. Some cancers are preceded by an inflammatory reaction in the body [25,26]. NF-κB signaling plays an important role in the control of cell growth, apoptosis, stress response, and inflammation. In contrast, lymphoid malignancies often harbor mutations that lead to aberrant activation of NF-κB signaling, and such malignancies can arise from all cellular stages of mature B-cell development. Several studies have shown that different NF-κB pathways and subunits have significant roles in the pathogenicity of lymphoma subtypes and myeloma. This provides us with an idea: by understanding the unique biological role of NF-κB in tumor precursor cells, we can use it to develop new targeted therapeutic drugs to inhibit the abnormal activation of NF-κB pathway to achieve the purpose of treatment [27,28,29]. According to the results of this study, we can conclude that C**4** can significantly stimulate the NF-kB inflammatory pathways and their targets and inhibit the Nrf2/HO-1 pathway to regulate intracellular redox balance, suggesting that it can be developed as a drug or functional food for more chronic diseases. Wang et al. have found that the glycosidic ligand on C**3** of *Camellia* seed is the main source of its anticancer activity [30]. During the experiment, we also studied C**3** and found that C**3** also has antitumor properties and promotes cell apoptosis activity, but that it is not as effective as C**4**, which is consistent with Wang’s findings. This implies that C**3** also has the potential to act as an anticancer drug or functional food. As a newly discovered compound, C**1** has not been studied in its physiological function or effect for the time being. We plan to screen its active components and functions with the help of multi-omics technology.

It is worth mentioning that, during the experiment, we found that C**4** can reduce the activity of Nrf2, and the effect was significant. This implies that C**4** targeted the Nrf2 signaling pathway to promote tumor cell apoptosis and has the potential to develop as an Nrf2 inhibitor to anticancer. The present study supplied meaningful molecular data to promote the application of C**4** in cancer drug development. Our study provided a new direction for this material, which is widely used in agriculture and in the food industry, as well as in new drug discovery.

## Figures and Tables

**Figure 1 antioxidants-12-00007-f001:**
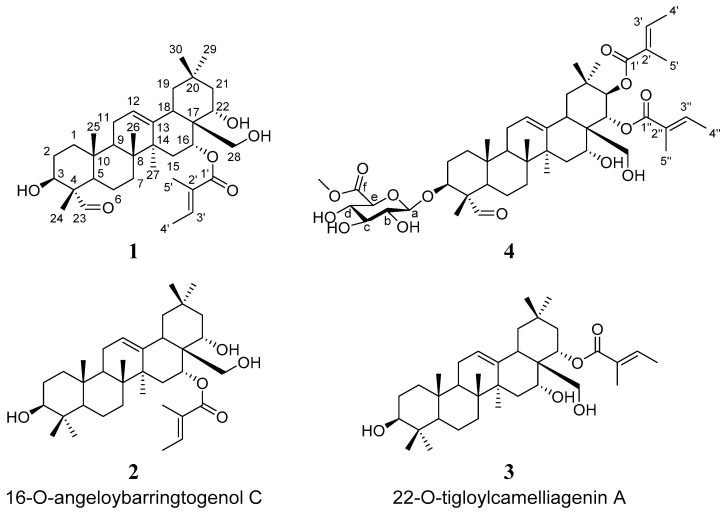
The structures of the theasaponin derivatives **1**−**4** from camellia seed cake.

**Figure 2 antioxidants-12-00007-f002:**
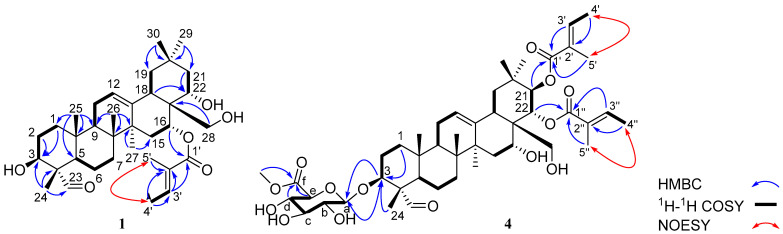
Key HMBC, ^1^H−^1^H COSY correlations of compounds **1** and **4**.

**Figure 3 antioxidants-12-00007-f003:**
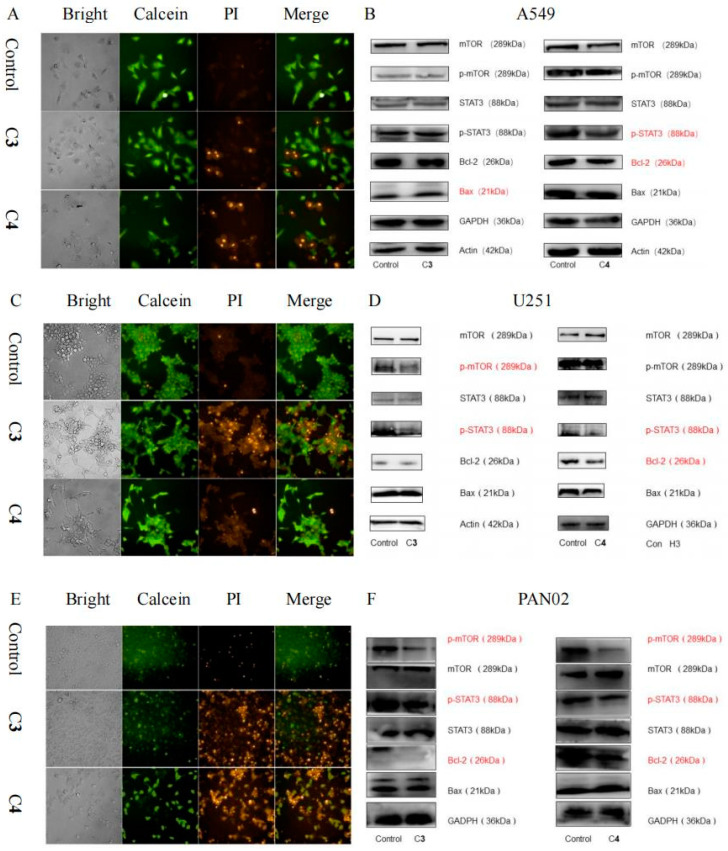
Effects of compounds **3** and **4** on the viability of three typical tumor cell lines. (**A**) Calcein AM/PI assay was used to detect the cytotoxicity of compounds **3** and **4** in A549 cells. (**B**) Compound **3** promotes apoptosis by up-regulating Bax, and compound **4** promotes apoptosis by down-regulating p-STAT3 and Bcl-2 in A549 cells. (**C**) Calcein AM/PI assay was used to detect the cytotoxicity of compounds **3** and **4** in U251 cells. (**D**) Compound **3** promotes apoptosis by down-regulating p-mTOR and p-STAT3, and compound **4** promotes apoptosis by down-regulating p-STAT3 and Bcl-2 in U251 cells. (**E**) Calcein AM/PI assay was used to detect the cytotoxicity of compounds **3** and **4** in PAN02 cells. (**F**) Compounds **3** and **4** promote apoptosis by down-regulating p-mTOR, p-STAT3, and Bcl-2 in PAN02 cells.

**Figure 4 antioxidants-12-00007-f004:**
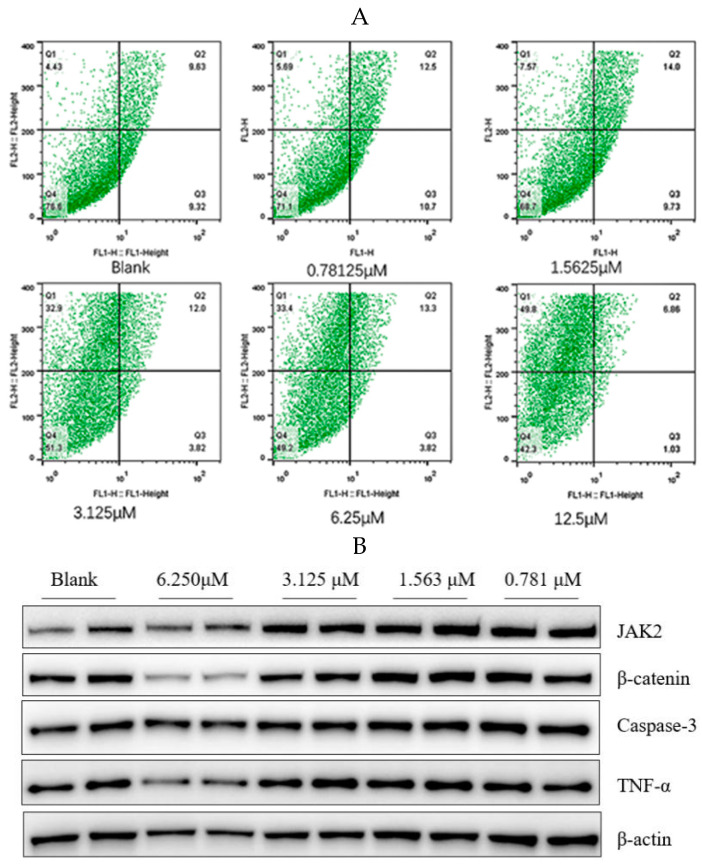
The dose-dependent antitumor activity of compound **4** in HepG2 cells. (**A**) The effect of compound **4** on apoptosis of HepG2 cells by flow cytometry. (**B**) The effect of compound **4** on typical protein expressions of apoptosis-related signaling pathways in HepG2 cells. * *p* < 0.05.

**Figure 5 antioxidants-12-00007-f005:**
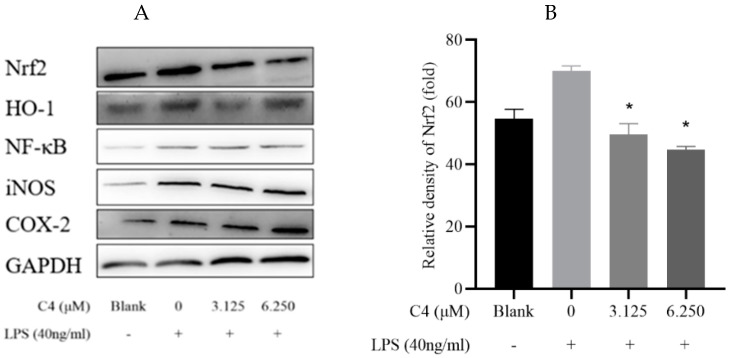
The expression of C4 on Nrf2 and NF-κB pathways in LPS-induced HepG2 cells. (**A**) protein bands; (**B**) Nrf2; (**C**) HO-1; (**D**) NF-κB; (**E**) iNOS; (**F**) COX-2. LPS + 3.125 μM, LPS + 6.250 μM vs. LPS, * *p* < 0.05.

**Table 1 antioxidants-12-00007-t001:** NMR spectroscopic data of **1**−**4** in CDCl_3_ (*δ* in ppm, *J* in Hz, 500 MHz for ^1^H NMR, 125 MHz for ^13^C NMR).

	1		4		2		3	
	*δ* _H_	*δ* _C_	*δ* _H_	*δ* _C_	*δ* _H_	*δ* _C_	*δ* _H_	*δ* _C_
1	1.69 m, 1.05 m	38.4	1.68 m, 1.05 m	38.3	1.64 m, 0.98 m	38.8	1.64 m, 1.01 m	38.8
2	1.71 m	26.2	1.80 m	20.5	1.61 m	27.3	1.59 m	27.3
3	3.75 dd (11.4, 4.6)	72.0	3.78 m	82.5	3.20 dd (11.0, 4.7)	79.1	3.21 dd (11.0, 4.6)	79.1
4		55.3		55.2		38.9		38.9
5	1.26 m	48.4	1.26 m	48.4	0.71 d (11.0)	55.4	0.73 d (11.3)	55.3
6	1.53 m, 0.98 m	20.8	1.51 m, 0.93 m	20.7	1.55 m, 1.38 m	18.4	1.54 m, 1.35 m	18.4
7	1.53 m, 1.26 m	32.1	1.60 m, 1.38 m	33.6	1.53 m, 1.32 m	32.8	1.54 m, 1.29 m	32.9
8		40.3		40.2		40.0		39.9
9	1.71 m	46.8	1.72 m	46.5	1.64 m	46.9	1.61 m	46.7
10		36.0		35.8		37.0		37.0
11	1.90 m	23.5	1.91 m	23.6	1.90 m	23.5	1.87 m	23.6
12	5.31 t (3.9)	123.7	5.44 s	124.4	5.31 t (3.4)	124.1	5.38 t (3.3)	124.2
13		141.4		141.2		141.3		141.8
14		41.4		41.2		41.4		41.2
15	2.11 m, 1.36 m	31.8	1.56 m, 1.20 m	32.2	2.12 m, 1.40 m	31.6	1.73 m, 1.32 m	34.0
16	5.80 brs	71.5	3.92 m	69.6	5.83 brs	71.7	4.02 brs	71.1
17		43.9		47.9		43.9		44.9
18	1.99 m	42.0	2.70 m	39.4	1.96 dd (14.0, 3.9)	42.0	2.56 dd (14.1, 4.5)	40.5
19	2.12 m, 1.12 m	46.4	2.57, 1.24	46.5	2.14 m, 1.10 m	46.4	2.31 m, 1.15 m	46.9
20		31.6		36.0		31.9		31.7
21	1.39 m, 1.15 m	44.1	5.86 d (10.0)	77.6	1.42 m, 1.18 d (12.7)	44.1	2.08 m, 1.55 m	42.7
22	3.89 dd (12.8, 5.4)	76.6	5.38 d (10.0)	73.3	3.89 dd (12.8, 5.3)	76.8	5.46 dd (12.4, 5.9)	72.3
23	9.37 s	207.3	9.40 s	208.1	0.97 s	28.2	0.98 s	28.2
24	1.05 m	9.1	1.10 s	10.1	0.93 s	15.7	0.92 s	15.7
25	0.98 s	16.0	0.97 s	16.0	0.77 s	15.7	0.77 s	15.7
26	0.93 s	16.7	0.89 s	16.9	0.93 s	16.7	0.89 s	16.9
27	1.40 s	27.3	1.45 s	27.2	1.40 s	27.3	1.42 s	27.2
28	3.64 d (11.3), 3.38 d (11.3)	73.2	3.24 m, 2.87 m	63.7	3.67 d (11.4), 3.39 d (11.4)	73.4	3.27 d (11.4), 2.97 d (11.4)	64.6
29	0.85 s	33.1	0.89 s	29.2	0.85 s	33.2	0.94 s	33.1
30	0.93 s	24.9	1.07 s	19.8	0.93 s	24.9	1.03 s	24.7
1′		169.5		167.9		169.6		169.0
2′		129.1		128.1		129.1		127.7
3′	6.91 q (6.8)	139.2	5.98 m	137.7	6.93 qd (7.0, 1.1)	139.2	6.11 qd (7.2, 1.4)	139.2
4′	1.82 d (6.8)	14.7	1.80 brs	20.5	1.82 dd (7.0, 1.1)	14.7	1.99 dd (7.2, 1.4)	16.0
5′	1.87 s	12.5	1.89 s	15.9	1.87, brs	12.5	1.89 t (1.4)	20.8
1″				169.4				
2″				127.3				
3″			6.07 d (7.3)	140.0				
4″			1.80 brs	20.6				
5″			1.93 s	16.0				
Gal								
a			4.27 d (7.5)	104.1				
b			3.33 m	73.3				
c			3.50 m	75.5				
d			3.66 m	71.4				
e			3.87 m	74.7				
f				169.9				
f-Me			3.80 s	52.9				

**Table 2 antioxidants-12-00007-t002:** The cell cytotoxicity of the isolates against human tumor cell lines.

	ED_50_ (μM)				
	1	2	3	4	DDP*^a^*
Huh-7	35.6	148.9	101.0	11.3	55.6
HepG2	43.0	115.6	10.8	1.5	8.1
Hela	13.8	78.3	21.2	4.6	9.5
A549	17.6	332.6	12.4	7.9	14.4
SGC7901	6.8	142.8	20.3	4.1	5.6
293T			3.314	2.195	23.43
U251			2.431	3.851	18.04
PAN02			6.396	4.931	6.225
MCF7			7.881	1.805	27.34

DDP*^a^* was used as a positive control.

## Data Availability

The data are contained within the article and Appendix A.

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
