# Peer review of "Discovery of New Triterpenoids Extracted from Camellia oleifera Seed Cake and the Molecular Mechanism Underlying Their Antitumor Activity"

_antioxidants, 2022, doi:10.3390/antiox12010007_

Round 1
Reviewer 1 Report
The manuscript by Wu and colleagues describe the effect of new triterpenoids from Camellia oleifera on cancer cell lines.
Comments
The effect of the compounds must be tested on normal cells, e.g. fibroblasts, as well.
Ln 137, please give information about the binding buffer.
Ln 256, delete the word “typical”.
Table 2, the effect of 1 and 2 on four cell lines is missing. As mentioned above, the effect on normal cells is missing, as well.
Fig. 3: Some of the blots are of low quality. Please provide better ones. Furthermore, in some cases the changes mentioned in the text are not reflected in the figure. Please provide densitometric analysis.
Ln 309: how are mTOR and STAT3 affect apoptosis in this cell system. Are these parallel phenomena or are they mechanistically linked? More experiments are needed to prove this connection.
Fig. 5A: Details in x-axis are missing.
General comment: please explain how C4 exerts a anti-tumor activity while it stimulates proliferation via the TNF-α/NF-kB pathway.
Author Response
Response to comments 1:
1. The effect of the compounds must be tested on normal cells, e.g. fibroblasts, as well.
Response: Thank you for your gentle reminder, and the 293T cells are normal cells for control as shown in Paragraph 3.2 and Table 2.
2. Ln 137, please give information about the binding buffer.
Response: Thanks for your advice. As suggested, we have added the detail information about the binding buffer as “Cells were cultured in 12-well culture paltes with different treatments for 4 h, then washed twice with cold PBS (Beyotime Biotechnology Co. Ltd., China) and resus-pended in binding buffer (Beyotime Biotechnology Co. Ltd., China) to a concentration of 1×106 cells/mL.”
3. Ln 256, delete the word “typical”.
Response: Thank you for your help. We have deleted the word “typical” accordingly.
4. Table 2, the effect of 1 and 2 on four cell lines is missing. As mentioned above, the effect on normal cells is missing, as well.
Response: Thank you for your comment. Paragraph 3.2, as shown in Table 2, 293T cells were normal cells. Considering the cytotoxic activities of the tested theasaponin derivatives and the fact that C3 and C4 have stronger antitumor capabilities than C1 and C2, we further investigated the effect of C3 and C4 on the expressions of proteins belonging to signaling pathways related to survival, proliferation, migration and invasion in cells.
5. Some of the blots are of low quality. Please provide better ones. Furthermore, in some cases the changes mentioned in the text are not reflected in the figure. Please provide densitometry analysis.
Response: Thank you for your suggestion, and we have carefully replaced all the blots of low quality to better ones accordingly.
6. Ln 309: how are mTOR and STAT3 affect apoptosis in this cell system. Are these parallel phenomena or are they mechanistically linked? More experiments are needed to prove this connection.
Response: Thank you for your question. Several studies have found that cell injury activates the mTOR/STAT3 signaling pathway, thereby reducing cell apoptosis, which means mTOR is an important signal transduction molecule. When mTOR is activated by external stimuli, the expression and activity of downstream signaling molecules in the pathway will be increased, which can not only promote cell growth, proliferation and differentiation, but also regulate cell apoptosis. STAT3 can be activated by many cytokines and growth factors and is readily phosphorylated by the activated mTOR kinase. Activated STAT3 plays an important role in the control of cell growth, proliferation, differentiation, and apoptosis. Therefore, these two has crosstalk.
7. 5A: Details in x-axis are missing.
Response: Thank you for your comment, and the details in x-axis have been supplemented.
8. General comment: please explain how C4 exerts a anti-tumor activity while it stimulates proliferation via the TNF-α/NF-kB pathway.
Response: Thank you for your comment. The results suggested that C4 may trigger apoptosis through Bcl-2/Caspase-3 and JAK2/STAT3 pathways, and stimulate cell proliferation via NF-kB/iNOS/COX-2 pathway. Normally, when cells are stimulated by existential threat, the TNF-α/NF-kB pathway may be activated to promote cell proliferation and confront death or apoptosis.
Reviewer 2 Report
Manuscript No. antioxidants-2041883
„Discovery of new triterpenoids extracted from Camellia oleifera seed cake and molecular mechanism study on their anti-tumor activity” for Antioxidants
Comments:
1. Abstract. Line 33. Please use the standard cisplatinum abbreviation as CDDP.
2. Introduction. Lines 79-85 are more suited to the results section. The introduction is intended to be a paragraph that introduces the reader to the topic.
3. Paragraph 2.4. First, please clearly describe how the tested substances were prepared for in vitro cell culture analyses. What was diluted in, what was the stock solution, what working concentrations were used, etc. Secondly, please provide the data of the cell cultures (tissue bank numbers), a one-sentence description of what kind of cultures were used (normal , tumor), what tissues the cells come from, etc. Thirdly, please describe the culture conditions (media, analysis times, etc.).
4. Paragraph 2.5. Line 135. Please describe the experiment in detail. What density of cells was used, what concentrations of substances were incubated with.
5. Paragraph 2.6. Lines 143-144. I understand that the Authors have entered culture medium, but what are 1% double antibodies means? Did the authors mean the antibiotics used? So it have to be written what antibiotics and in what concentrations.
6. Paragraph 2.7. Line 155. I suppose it was the number of cells/mL. In addition, I am asking for more detailed data on the antibodies used.
7. Table 2. Why in the text the Authors write about five cultures, but in the table they mark as many as nine?
8. Paragraph 3.4. Unfortunately, the analysis of the results from the cytometer is very poor. The authors basically did not show these results. Please elaborate them carefully.
9. Why were different experiments done on different cultures? This causes serious confusion. If the Authors adopted five cultures, all results should also be presented on these cells. Otherwise, the choice of these cultures misses the point.
10. The discussion is very superficial. In principle, the obtained results have not been explained and what they may mean. This chapter should be re-written.
11. Please also standardize the font size in the text.
Author Response
1. Line 33. Please use the standard cisplatinum abbreviation as CDDP.
Response: Thanks a lot for your gentle reminder, and we have used the standard cisplatinum abbreviation as CDDP.
2. Lines 79-85 are more suited to the results section. The introduction is intended to be a paragraph that introduces the reader to the topic.
Response: Thanks for your advice, and we have carefully revised the introduction section accordingly.
3. Paragraph 2.4. First, please clearly describe how the tested substances were prepared for in vitro cell culture analyses. What was diluted in, what was the stock solution, what working concentrations were used, etc. Secondly, please provide the data of the cell cultures (tissue bank numbers), a one-sentence description of what kind of cultures were used (normal , tumor), what tissues the cells come from, etc. Thirdly, please describe the culture conditions (media, analysis times, etc.).
Response: Thank you for your good suggestion and the sentence has been revised as shown in the Paragraph 2.4.
4. Paragraph 2.5. Line 135. Please describe the experiment in detail. What density of cells was used, what concentrations of substances were incubated with.
Response: It is a good suggestion and we have made corresponding modifications in the manuscript as “Cellswere cultured in 12-well culture paltes with different treatments for 4 h, then washed twice with cold PBS (Beyotime Biotechnology Co. Ltd., China) and resuspended in binding buffer (Beyotime Biotechnology Co. Ltd., China) to a concentration of 1×106 cells/mL.”
5. Paragraph 2.6. Lines 143-144. I understand that the Authors have entered culture medium, but what are 1% double antibodies means? Did the authors mean the antibiotics used? So it have to be written what antibiotics and in what concentrations.
Response: Many thanks for your advice and we have revised them accordingly as “Cells were cultured in DMEM medium containing 10% FBS (Beyotime Biotech-nology Co. Ltd., China) and 1% double antibiotics(100 U/mL penicillin, 100 mg/mL streptomycin, Gibco, USA), then washed 5mL PBS.”
6. Paragraph 2.7. Line 155. I suppose it was the number of cells/mL. In addition, I am asking for more detailed data on the antibodies used.
Response: Thank you for your reminder, and we have added the detailed information required accordingly as shown in the Paragraph 2.7.
7. Table 2. Why in the text the Authors write about five cultures, but in the table they mark as many as nine?
Response: Thank you for your comment.Considering the cytotoxic activities of the tested theasaponin derivatives and the fact that C3 and C4 have stronger antitumor capabilities than C1 and C2, we further investigated the effect of C3 and C4 on the expressions of proteins belonging to signaling pathways related to survival, proliferation, migration and invasion in cells.
8. Paragraph 3.4. Unfortunately, the analysis of the results from the cytometer is very poor. The authors basically did not show these results. Please elaborate them carefully.
Response: Thank you for the reminder, and we have elaborated the result statement in the manuscript as shown in Paragraph 3.4.
9. Why were different experiments done on different cultures? This causes serious confusion. If the Authors adopted five cultures, all results should also be presented on these cells. Otherwise, the choice of these cultures misses the point.
Response: Thank you for the question. We chose different cultures for different cells since cells have their specific nutritional needs and culture parameters, and this difference cannot affect the antitumor property assay result of different cells.
10. The discussion is very superficial. In principle, the obtained results have not been explained and what they may mean. This chapter should be re-written.
Response: Thank you for your reminder, and we have added some in-depth discussion statement as shown in the last section.
11. Please also standardize the font size in the text.
Response: Thank you for the reminder. We have standardized the font size in the whole manuscript.
Round 2
Reviewer 1 Report
The authors have replied to most of the reviewer's comments. However, some of the issues are still unanswered:
Binding buffer: please provide the appropriate description.
For the role of mTOR and STAT3, information from the literature is not adequate. The authors must check the role of these components in their own system by using appropriate inhibitors, siRNA or similar approaches.
The authors view, as shown in the discussion, that C4 provokes both apoptosis and proliferation is not clear. Please, refer only to the effect on apoptosis.
293T cells are not normal, as they carry SV40 T-antigen. Please use a normal cell strain instead, as asked in the previous report.
Author Response
The authors have replied to most of the reviewer's comments. However, some of the issues are still unanswered:
1. Binding buffer: please provide the appropriate description.
Response: Thanks a lot for your gentle reminder. As suggested, we have added the detail information about the binding buffer in Materials section as following:
“According to the instruction of the Annexin V-FITC/PI kit (Beyotime Biotechnology Co. Ltd., China), resuspended in Annexin binding buffer (10 mM HEPES, 140 mM NaCl, 2.5 mM CaCl2 and pH 7.4) to a concentration of 1×106 cells/mL.”
2. For the role of mTOR and STAT3, information from the literature is not adequate. The authors must check the role of these components in their own system by using appropriate inhibitors, siRNA or similar approaches.
Response: Thanks for your advice, and we clarify the role of mTOR and STAT3 in the conclusion section as following.
“In the present study, Compound 3 was by down-regulating p-mTOR and p-STAT3, and C4 was by down-regulating p-STAT3 and Bcl-2 in U251 cells. Compound 3 and Compound 4 were by down-regulating p-mTOR, p-STAT3, and Bcl-2 in PAN02 cells. Cell injury activates the mTOR/STAT3 signaling pathway, thereby reducing cell apoptosis, which means mTOR is an important signal transduction molecule. When mTOR is activated by external stimuli, the expression and activity of downstream signaling molecules in the pathway will be increased, which can not only promote cell growth, proliferation and differentiation, but also regulate cell apoptosis. STAT3 can be activated by many cytokines and growth factors and is readily phosphorylated by the activated mTOR kinase. Activated STAT3 plays an important role in the control of cell growth, proliferation, differentiation, and apoptosis. Therefore, mTOR and STAT3 has crosstalk. In the future, we will conduct further studies to examine the effects of these components on mTOR/STAT3 signaling pathway by using appropriate inhibitors, siRNA or similar approaches.
In the near future, we are sure to conduct further study to investigate the in-depth molecular mechanism underlying the effects of these components on mTOR/STAT3 signaling pathway by using appropriate inhibitors, including siRNA, CRISPR/Cas9 gene edit or similar approaches.
3. The authors view, as shown in the discussion, that C4 provokes both apoptosis and proliferation is not clear. Please, refer only to the effect on apoptosis.
Response: It is a good suggestion and we have made corresponding modifications as shown in Discussion section of the manuscript accordingly and seriously, many thanks!
4. 293T cells are not normal, as they carry SV40 T-antigen. Please use a normal cell strain instead, as asked in the previous report.
Response: Thank you for your suggestion. Although 293T cells carry SV40 T-antigen, but the SV40 T-antigen did not have any effect on the result of our present study. According to several related papers published recently, 293T cell lines was considered as noncancerous cell lines and were often used as normal control, which is usually used as referee cell in cancer research. These references are listed below:
(1) Koçak N, Dönmez H, Yildirim İH. Effects of melatonin on apoptosis and cell differentiation in MCF-7 derived cancer stem cells. Cell Mol Biol (Noisy-le-grand). 2018 Sep 30;64(12):56-61.
(2) Khazraei-Moradian S, Ganjalikhani-Hakemi M, Andalib A, Yazdani R, Arasteh J, Kardar GA. The Effect of Licorice Protein Fractions on Proliferation and Apoptosis of Gastrointestinal Cancer Cell Lines. Nutr Cancer. 2017 Feb-Mar;69(2):330-339.
Reviewer 2 Report
The manuscript is corrected.
Author Response
1. The manuscript is corrected.
Response: Thank you for your comments!